

# When and where? Day-night alterations in wild boar space use captured by a generalized additive mixed model

Martijn Bollen[1,2,3], Jim Casaer[2], Thomas Neyens[3,4] and Natalie Beenaerts[1]

[1] Centre for Environmental Sciences, Hasselt University, Hasselt, Flanders, Belgium
[2] Research Institute for Nature and Forest (INBO), Brussels, Brussels, Belgium
[3] Data Science Institute, Hasselt University, Hasselt, Flanders, Belgium
[4] Leuven Biostatistics and statistical Bioinformatics Centre, University of Leuven, Leuven, Flanders, Belgium

## ABSTRACT

Wild boar (*Sus scrofa*), an abundant species across Europe, is often subjected to management in agro-ecosystems in order to control population size, or to scare them away from agricultural fields to safeguard crop yields. Wild boar management can benefit from a better understanding on changes in its space use across the diel cycle (*i.e.*, diel space use) in relation to variable hunting pressures or other factors. Here, we estimate wild boar diel space use in an agro-ecosystem in central Belgium during four consecutive "growing seasons" (*i.e.*, April–September). To achieve this, we fit generalized additive mixed models (GAMMs) to camera trap data of wild boar aggregated over 1-h periods. Our results reveal that wild boar are predominantly nocturnal in all of the hunting management zones in Meerdaal, with activity peaks around sunrise and sunset. Hunting events in our study area tend to take place around sunrise and sunset, while non-lethal human activities occur during sunlight hours. Our GAMM reveals that wild boar use different areas throughout the diel cycle. During the day, wild boar utilized areas in the centre of the forest, possibly to avoid human activities during daytime. During the night, they foraged near (or in) agricultural fields. A *post hoc* comparison of space use maps of wild boar in Meerdaal revealed that their diurnal and nocturnal space use were uncorrelated. We did not find sufficient evidence to prove that wild boar spatiotemporally avoid hunters. Finally, our work reveals the potential of GAMMs to model variation in space across 24-h periods from camera trap data, an application that will be useful to address a range of ecological questions. However, to test the robustness of this approach we advise that it should be compared against telemetry-based methods to derive diel space use.

## INTRODUCTION

Wild boar population densities are increasing across Europe (*Carpio, Apollonio & Acevedo, 2021*; *Massei et al., 2015*). Consequently, human-wild boar interactions are becoming more frequent, leading to both positive and negative encounters. From an economical perspective, damage to agricultural crops is one of the most important impacts of wild

Corresponding author
Martijn Bollen,
martijn.bollen@uhasselt.be

boar, with economic losses amounting to hundreds of thousands of Euros per year in several European countries (*Lombardini, Meriggi & Fozzi, 2016*). Wild boar damage to croplands primarily occurs during the growing season, when crops are ripe (*Herrero et al., 2006*; *Kramer et al., 2022*). Crops that provide both shelter and feeding opportunities, such as maize or wheat, appear to be particularly at risk (*Kramer et al., 2022*). In an attempt to safeguard crop yields during the growing season, wild boar are typically under moderate to high hunting pressure. Yet, the current hunting practices across Europe appear to be ineffective at controlling wild boar population size and mitigating their impact on agricultural crops (*Massei et al., 2015*). Reducing wild boar population size has been achieved most effectively through coordinated and adaptive strategies by wildlife professionals (*Treichler et al., 2023*). In the absence of this type of eradication programs, increasing the effectiveness of recreational hunting at mitigating crop loss caused by wild boar is essential.

An appealing strategy is to create zones of differential hunting pressure (*i.e.,* creating a "landscape of fear"), with the highest hunting pressure near agricultural fields in order to prevent wild boar from using them as a foraging ground (*Tolon et al., 2009*). Indeed, it appears that wild boar shift their space use in response to hunting in some cases (*Colomer et al., 2021*; *Tolon et al., 2009*), but not in others (*Brogi et al., 2020*; *Reinke et al., 2021*; *Wevers et al., 2020*). These conflicting results may be partly explained by differences in the number of hunting posts occupied at the same time, which is a key determinant of hunting success and possibly also in the successful modulation of wild boar space use (*Quirós-Fernández et al., 2017*; *Vajas et al., 2020*). Moreover, wild boar are known to shift their activities towards increased nocturnality in human-dominated landscapes or in response to hunting (*Johann et al., 2020*; *Keuling, Stier & Roth, 2008b*; *Ohashi et al., 2013*; *Podgórski et al., 2013*). This allows them to avoid places, such as agricultural crops, associated with high risk when humans are active, while still using them at night when human activity is low (*Kramer et al., 2022*; *Stillfried et al., 2017*). Because of this trade-off, preventing crop damage by wild boar requires better insight into how these animals respond to differential hunting pressure throughout the diel cycle.

Investigating the influence of hunting on diel variations in the space use of wild boars, requires proper statistical tools. While species distribution models are widely applied to study biotic and abiotic factors that influence animal space use, including the impacts of hunting (*Di Bitetti et al., 2008*; *Guisan & Thuiller, 2005*), they typically obscure any changes in spatial patterns that occur within 24-h periods. This is because species distribution models require that the user defines a time period (*i.e.,* the time of a single "survey" or "temporal replicate") over which species records are aggregated, which is typically 24-h or more to increase the probability of detection (*Bassing et al., 2023*; *Caruso et al., 2018*; *Crunchant et al., 2020*; *Rich et al., 2017*; *Shannon, Lewis & Gerber, 2014*). Several recent studies have considered space use in combination with diel activity. These studies have used specific forms of occupancy models (*Kellner et al., 2022*; *Rivera et al., 2022*), a spatial capture-recapture model (*Distiller et al., 2020*), a MAXENT model (*Campanella et al., 2019*), a model based on encounter rates (*Ait Kaci Azzou et al., 2021*) and resource selection models (*Gallo et al., 2022*; *Kohl et al., 2018*). However, most of them treat space use and

diel activity separately, such that space use is fixed across the diel cycle (*Ait Kaci Azzou et al., 2021*; *Distiller et al., 2020*; *Kellner et al., 2022*). Others have allowed spatial patterns to change throughout the diel cycle but only as a function of measurable covariates and between broad time categories (*e.g.,* night, day, dawn, dusk) (*Campanella et al., 2019*; *Gallo et al., 2022*; *Rivera et al., 2022*). However, if species adapt their space use patterns gradually across time or if covariate information is missing, this may result in critical loss of information regarding their diel space use. Thus, being able to obtain diel space use from camera trap data where the spatial pattern can change in continuous time possibly without additional covariate information, would extend the capabilities of the current methodology. Therefore, we applied a generalized additive mixed models (GAMM), which allows the spatial pattern to change in continuous time while also accounting for random effects. The major drawback of GAMMs is that they do not account for imperfect detections, as is possible in other frameworks (*Ait Kaci Azzou et al., 2021*; *Distiller et al., 2020*; *Kellner et al., 2022*; *Rivera et al., 2022*). In addition, the flexibility of GAMMs can also make them prone to overfitting changes in space use patterns.

In general, the objective of our study is to bridge the knowledge gap related to diel space use of wild boar in relation to hunting pressure in an agro-ecosystem. In relation to our study area, we are interested in evaluating whether the local hunting efforts are sufficient to trap wild boar in the centre of the forest during the time when crops are growing to safeguard crop yields. Thus, we only included data from six months growing seasons (April–September), when crops are ripe, in our study. We hypothesize that wild boar are mainly nocturnal (H1) and that their space use pattern changes throughout the diel cycle (H2). Specifically, we expect wild boar to rest throughout the day in areas distant from non-lethal human disturbance (*i.e.,* utilize the centre of the forest) (H3). During the night, we expect that they utilize a larger area, including sites near agricultural fields (H4). Finally, we hypothesize that hunting influences the diel space use of wild boar (H5).

## MATERIALS & METHODS

### Study area

The study area (longitude: 4.650°W– 4.750°W; latitude: 50.788°N–50.824°N) is situated in a Natura 2000 reserve called "Meerdaal" in central Belgium (Fig. 1A). Meerdaal has altitudes ranging from 35 to 103 m above sea level and is characterised by locally steep slopes. The study area has a cool temperate and moist climate, with a mean annual temperature of 11°C and 773.2 mm mean annual rainfall (*KMI, 2021*). It has a total surface area of ~16 km$^2$, consisting of a mosaic of coniferous (mainly *Pinus sylvestris*) and broad-leaved (mainly *Quercus spp.*, *Fagus sylvatica* and *Carpinus betulus*) forest stands. Acorns and beechnuts represent the dominant mast species and are mostly available from October through December, homogenously distributed throughout the area. The forested area in Meerdaal is surrounded by a rich mosaic of croplands, with crops growing predominantly during April–September. In the context of an European observatory of wildlife project by ENETWILD, wild boar density in and around Meerdaal has been estimated at 7.88 ± 3.50 individuals/km$^2$ using a random encounter model (*Guerrasio et al., 2023*; *Rowcliffe et*

*al., 2008*). Hunting in Meerdaal, except for drive hunts, is restricted to fixed locations (*i.e.,* elevated hunting posts), and can only take place between 19:00 and 9:00 during Daylight Saving Time, and between 16:00 and 10:00 during Winter Time. The study area is subdivided into three hunting management zones with different intensities of hunting pressure. In the year-round hunting zone ('HY'; ~9 km$^2$), hunting of wild boar is allowed during the entire year. In the winter hunting zone ('HW'; ~4 km$^2$), hunting is restricted to November through March. In the hunting-restricted core zone ('C'; ~2 km$^2$), hunting is prohibited year round, with the exception of one or two silent drive hunts and four group hunts from elevated hunting posts during the winter. Note that during the study period (April–September), the central zones (HW and C) are, in principle, both free of hunting (Fig. 1B). For details on the hunting pressure in these zones, we refer readers to Table S1.

## Wild boar and human activity

As part of a larger monitoring framework, a subset of 13 cameras (Reconyx Hyperfire HC600; detection radius $r = 15$ m and angle of view $\theta = 42°$) has been deployed in Meerdaal since March 2018 (Fig. 1B). Cameras were placed at the centre of a subset of 250 m $\times$ 250 m grid cells (0.0625 km$^2$) that were selected from a grid overlaying the study area following a spatially-balanced sampling scheme (*Stevens & Olsen, 2004*). All cameras were relocated monthly to a new grid cell location. Annually, the same set of grid cells was visited twice: once during the summer (April–September) and a second time during the winter (October–March). All cameras were mounted ~50 cm above ground, facing north, on the tree nearest to the middle of the selected grid cell. This resulted in camera locations which were on average 242 m away from the closest hunting post (range: 9 m–925 m). None of the cameras were baited to lure animals, or placed along a trail to avoid bias from baiting and/or preferential sampling. Each camera trigger was followed by a sequence of ten consecutive photos, with a 0-s recovery time between triggers. We considered sequences (10 photos/trigger) to be independent if they were a least 60 min apart. We also assessed shorter times to independence (*i.e.,* 2 min and 30 min), which did not substantially change the findings of our study (results not shown). Non-independent sequences were aggregated and annotated as a single sequence of >10 photos. We considered each independent sequence to display an independent group of wild boar or humans and defined the raw counts as the number of unique individuals in these groups. Annotation was done using the Agouti software platform (http://www.agouti.eu). For our analysis, we only considered images from a six months growing seasons (April–September) of the years 2018 through 2021. During this period, all cameras remained operative (*i.e.,* no stolen cameras or defects). Because COVID-19 related lockdowns can strongly impact animal activity and space use, we chose to exclude data from the most stringent lockdown period in Belgium (18 March 2020–10 May 2020). This yielded a total of 9,542 24-h observation periods from 303 camera deployments. In this period, 1,085 independent groups of wild boar were captured (total count: 2,532, average group size: 2.33, range group size: 1–25) and 99 human sightings were recorded (total count:142, average group size: 1.43, range group size: 1–6). Wild boar and human counts $^*$ day$^{-1}$ $^*$ solar hour$^{-1}$, which were obtained by dividing raw counts per solar hour (details on solar hours are provided in the Statistical Analyses sub-section)

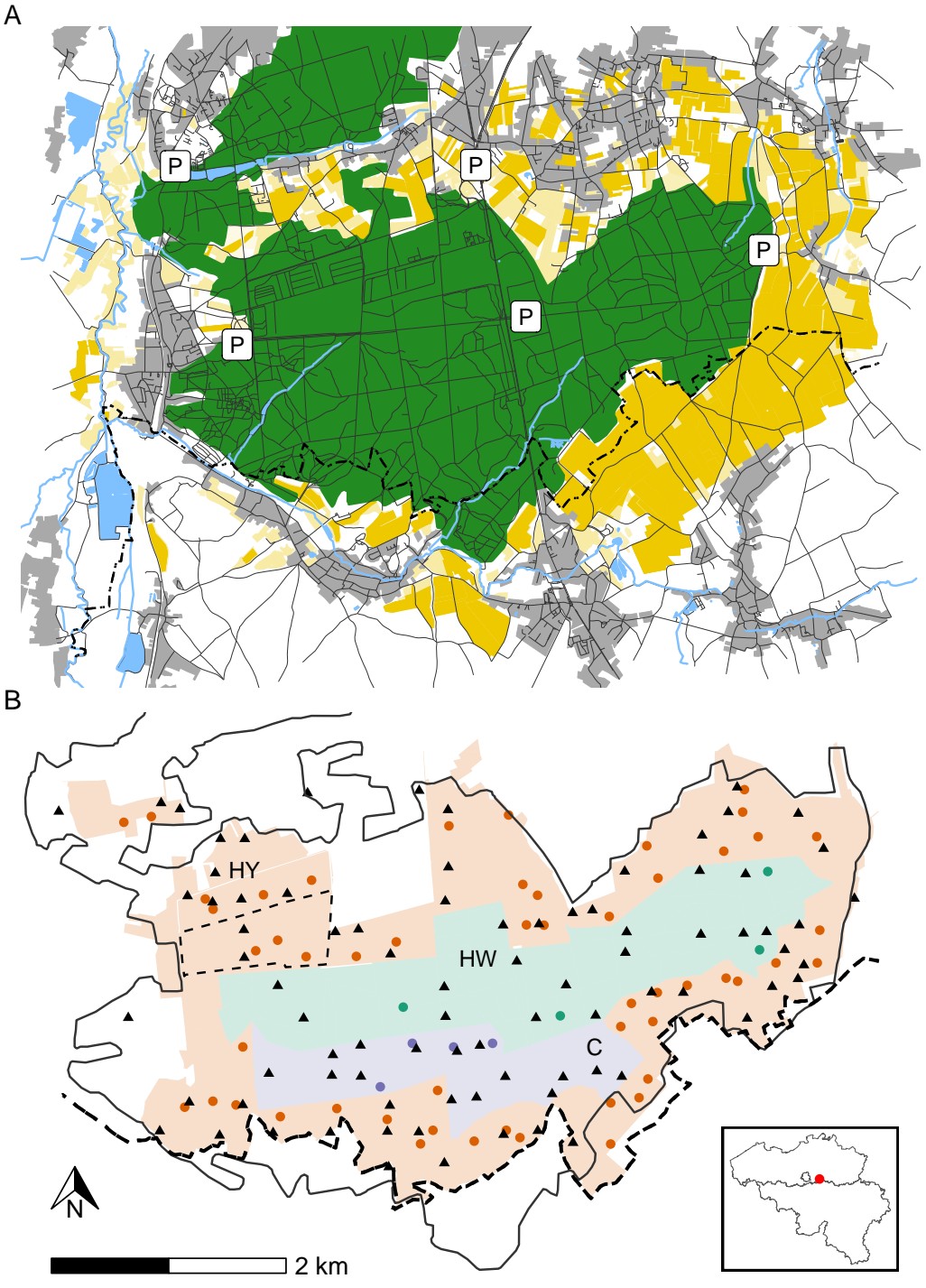

**Figure 1 Map of the study area illustrating major landscape types (A), management zones, locations of cameras and hunting posts (B).** Panel A: forests (green), agricultural fields (yellow), rivers, streams and waterbodies (blue), urban areas, roads or trails (grey). Panel B: Year-round hunting zone (HY—red), winter hunting zone (HW—green) and hunting-restricted core zone (C—blue). Camera locations (triangles) and elevated hunting posts (circles). The full black line marks the forest edge, while the dashed line indicates the administrative border between Flanders and Wallonia. The inset map shows the location of the study area in Belgium.

**Table 1 The number of wild boar, humans and hunters counted.** Counts represent the total number of individuals within each solar hour over the entire study period. In addition, expected counts per camera per day (count/cam. day) are represented for humans and wild boar. For hunters, expected counts per day are provided (count/day).

| Solar hour | Wild boar | | Humans | | Hunters | |
|---|---|---|---|---|---|---|
| | Count | Count/ cam. day | Count | Count/ cam. day | Count | Count/ day |
| $(0/24)\cdot 2\pi$ | 71 | 0.007 | 0 | 0.000 | 1 | 0.001 |
| $(1/24)\cdot 2\pi$ | 65 | 0.007 | 0 | 0.000 | 1 | 0.001 |
| $(2/24)\cdot 2\pi$ | 64 | 0.007 | 0 | 0.000 | 1 | 0.001 |
| $(3/24)\cdot 2\pi$ | 161 | 0.017 | 0 | 0.000 | 12 | 0.017 |
| $(4/24)\cdot 2\pi$ | 139 | 0.015 | 1 | <0.001 | 106 | 0.153 |
| $(5/24)\cdot 2\pi$ | 153 | 0.016 | 1 | <0.001 | 217 | 0.314 |
| $(6/24)\cdot 2\pi$ | 276 | 0.029 | 2 | <0.001 | 242 | 0.350 |
| $(7/24)\cdot 2\pi$ | 153 | 0.016 | 3 | <0.001 | 173 | 0.250 |
| $(8/24)\cdot 2\pi$ | 126 | 0.013 | 9 | 0.001 | 15 | 0.022 |
| $(9/24)\cdot 2\pi$ | 40 | 0.004 | 18 | 0.002 | 1 | 0.001 |
| $(10/24)\cdot 2\pi$ | 16 | 0.002 | 6 | 0.001 | 0 | 0.000 |
| $(11/24)\cdot 2\pi$ | 14 | 0.001 | 12 | 0.001 | 0 | 0.000 |
| $(12/24)\cdot 2\pi$ | 19 | 0.002 | 6 | 0.001 | 0 | 0.000 |
| $(13/24)\cdot 2\pi$ | 10 | 0.001 | 22 | 0.002 | 0 | 0.000 |
| $(14/24)\cdot 2\pi$ | 16 | 0.002 | 18 | 0.002 | 0 | 0.000 |
| $(15/24)\cdot 2\pi$ | 23 | 0.002 | 10 | 0.001 | 9 | 0.013 |
| $(16/24)\cdot 2\pi$ | 22 | 0.002 | 6 | 0.001 | 235 | 0.340 |
| $(17/24)\cdot 2\pi$ | 146 | 0.015 | 9 | 0.001 | 388 | 0.561 |
| $(18/24)\cdot 2\pi$ | 216 | 0.023 | 1 | <0.001 | 349 | 0.504 |
| $(19/24)\cdot 2\pi$ | 260 | 0.027 | 0 | 0.000 | 89 | 0.129 |
| $(20/24)\cdot 2\pi$ | 175 | 0.018 | 0 | 0.000 | 23 | 0.033 |
| $(21/24)\cdot 2\pi$ | 125 | 0.013 | 1 | <0.001 | 11 | 0.016 |
| $(22/24)\cdot 2\pi$ | 89 | 0.009 | 0 | 0.000 | 7 | 0.010 |
| $(23/24)\cdot 2\pi$ | 117 | 0.012 | 0 | 0.000 | 3 | 0.004 |
| Total | 2,496 | 0.262 | 125 | 0.02 | 1,883 | 2.721 |

through the total number of trapping days of all cameras combined (9,542) are presented in Table 1.

## Hunting effort

Within Meerdaal, it is mandatory for hunters to record their activities in a hunting diary. From 2018 through 2021, we have information from 3,460 different hunting events at 60 hunting posts (Fig. 1B), of which 1,131 occurred during the study period. After removing observations without information on the hunting effort (duration in hours) or hunting post used, we retained 1,114 records (98.5%). To reliably represent the total hunting effort in hunter activity patterns, we created "new" hunting records every 10 min between the start and end time of a hunting activity recorded in the diary. This yielded 8,868 time records for hunting activity, which we used to model the diel activity of hunters in Meerdaal. To model the spatiotemporal hunting pressure in Meerdaal, we only considered whether a

**Table 2  Candidate models for wild boar trapping rate.** The mathematical structure is presented together with selection criteria. The highest-ranking model is indicated in bold.

| Model | $log(\lambda_{ijt}) =$ | mdf | dev. expl. (%) | AIC | $\Delta$dev. expl. | $\Delta$AIC |
|---|---|---|---|---|---|---|
| M1 | $\beta_0 + f_1(t)$ | 4 | 0.101 | 16,219 | −0.250 | 580.47 |
| M2 | $(\beta_0 + \beta_{0,j}) + f_1(t)$ | 270 | 0.260 | 16,073 | −0.091 | 433.95 |
| M3 | $(\beta_0 + \beta_{0,j}) + f_1(t) + f_2(week(j))$ | 266 | 0.260 | 16,066 | −0.091 | 426.86 |
| M4 | $(\beta_0 + \beta_{0,j}) + f_1(t) + f_2(week(j)) + f_3(lon(i), lat(i))$ | 277 | 0.316 | 15,797 | −0.035 | 157.81 |
| **M5** | $(\boldsymbol{\beta_0} + \boldsymbol{\beta_{0,j}}) + \boldsymbol{f_1(t)} + \boldsymbol{f_2(week(j))} + \boldsymbol{f_3(t, lon(i), lat(i))}$ | **297** | **0.351** | **15,639** | **0.000** | **0.00** |
| M6 | $(\beta_0 + \beta_{0,j}) + \beta_1 Hunt_{jt} + f_1(t) + f_2(week(j)) + f_3(t, lon(i), lat(i))$ | 298 | 0.351 | 15,640 | 0.000 | 1.47 |

**Notes.**
mdf, model degrees of freedom; dev. expl. (%), percentage of deviance explained; AIC, Akaike information criterion.

hunter was present at a given hunting post during a specific time of the day (*i.e.,* solar hour). We obtained the hunter counts $^\star$ day$^{-1}$ $^\star$ solar hour$^{-1}$ by aggregating the number of hunters present across all hunting posts and days in the study period for each solar hour (Table 1).

## Statistical analysis
### Wild boar and human activity
Each unique camera deployment $i = 1, 2, \ldots, R$ produced pictures of wild boar and humans that were tagged with information on their coordinates $lon(i), lat(i)$ the survey day $j = 1, 2, \ldots, J_i$ and the "solar hour" of observation $t = 0, \frac{2\pi}{24}, \ldots, 2\pi$. We first obtained (continuous) solar times $t^*$ by mapping clock times to $[0, 2\pi]$ and anchoring these radian times to sunrise ($t_1 = \frac{\pi}{2}$) and sunset ($t_2 = \frac{3\pi}{2}$) on the day and location of the observation using the *SunTime()* function from the R package *overlap* (*Nouvellet et al., 2012*; *Ridout & Linkie, 2009*). This ensured that wild boar and human behavior was studied relative to standardized times (*i.e.,* solar events that are considered important regulators of cyclic patterns recurring each day) rather than exact clock times (*Nouvellet et al., 2012*; *Vazquez et al., 2019*). Second, we defined the lower bound of one of 24 evenly spaced intervals of $\frac{2\pi}{24}$ between 0 and $2\pi$ that holds the solar time $t^*$ as the (discrete) solar hour $t$.

To explore our data, we estimated wild boar and human activity patterns and overall activity levels using conventional methods for the three hunting management zones. More specifically, we fit a circular kernel density function $c(t^*)$ to solar times $t^*$ using the *fitact()* function from the R package *activity* (*Rowcliffe et al., 2014*). In order to obtain accurate density functions from this kernel estimator, a minimum of 100 time records is recommended (*Lashley et al., 2018*; *Rowcliffe et al., 2014*). For the management zones, we collected a total of 1,020 (HY), 304 (HW) and 559 (C) time records of wild boar during the study period. For humans we had access to 148 time records. Hence, we are confident that these activity patterns accurately represent the true underlying wild boar or human activity. The function *fitact()* also calculates the absolute overall activity levels as $\frac{1}{2\pi c_{max}}$ (*Rowcliffe et al., 2014*). To assesses differences in overall activity levels, we performed a Wald test on each pairwise comparison using the *compareAct()* function from *activity*. Finally, we identified the solar times $t^*$ at which the two strongest peaks (local maxima) in

$c(t^*)$ occur, by calculating the argmax locally. If wild boar are nocturnal (H1), we expect an activity pattern displaying sustained activity during nighttime and low activity during daytime. Moreover, activity peaks should not occur around sunrise ($\frac{\pi}{2}$) or sunset ($\frac{3\pi}{2}$), which is typical for crepuscular activity.

### Wild boar trapping rate

To obtain diel space use of wild boar, we adopted a GAMM, a type of regression model that allows the relationship between the outcome and one or more predictors to be smooth curves (*Hastie & Tibshirani, 1986*; *Wood, 2017*). We assumed that counts $y_{ijt}$ captured by camera $i$ on day $j$, resulting from aggregating all observations with solar hours $t$ follow a negative binomial distribution:

$$y_{ijt} \sim NegBin(\lambda_{ijt}, \theta),$$

with $\lambda_{ijt}$ the expected trapping rate (individuals $\star$ camera$^{-1}\star$ solar hour$^{-1}$ $\star$ day$^{-1}$) at camera $i$ on day $j$, for a given solar hour $t$ and $\theta$ an overdispersion parameter. We explicitly chose a negative binomial distribution because initial inspection of our data suggested that wild boar counts were overdispersed relative to a Poisson distribution, but we also explored the goodness-of-fit statistics from the latter. Note that a zero-inflated Poisson would be another sensible choice for our data, but this did not lead to convergence of our model. We modelled $\lambda_{ijt}$ in function of fixed and/or random effects using a log-link through a GAMM using the R package *mgcv* (*Wood, 2011*). We considered the following information to be used as fixed and/or random effects potentially affecting $\lambda_{ijt}$: solar time, survey day, week, longitude and latitude of the observation and the hunting effort on each solar hour of a given survey day. Using this information we evaluated six candidate models (Table 2). The remainder of this section describes the full model, including all the effects. For this model, the trapping rate $\lambda_{ijt}$ is expressed as:

$$log(\lambda_{ijt}) = (\beta_0 + \beta_{0,j}) + \beta_1 Hunt_{jt} + f_1(t) + f_2(week(j)) + f_3(t, lon(i), lat(i)),$$

where $\beta_0$ is a general intercept, $\beta_{0,j}$ represent random intercepts for each survey day and $\beta_1$ captures the effect of the total duration (in radians) of hunting on day $j$ at solar time $t$. The model also included two global smoothing terms, one for the solar times $f_1(t)$ and another for weeks of the year $f_2(week(j))$. Both were based on a cyclic cubic regression spline ('bs = cc' in *mgcv*), since solar and seasonal events are inherently periodic. Lastly, it included a 3d smoother for solar times, longitude and latitude $f_3(t, lon(i), lat(i))$, which is approximated by the superposition of three simpler basis functions $f_1(t)$, $f_{lon}(lon)$ and $f_{lat}(lat)$. In *mgcv*, this is done by taking the tensor products of these components using the function *te()*. For $f_{lon}(lon)$ and $f_{lat}(lat)$, we used thin plate regression splines ('bs = tp' in *mgcv*) because they are considered a general purpose spline (*Wood, 2003*). A grid search to determine the optimal number of knots $k$ based on the Akaike information criterion (AIC; *Akaike, 1974*) indicated that $k = 10$ was optimal. However, this yielded smooth functions that overfitted the data. Hence, we explored a progressively smaller number of knots until this overfitting behavior disappeared. Eventually, $k = 5$ was used for all terms. Note that the data $y_{ijt}$ was typically very sparse, which may lead to poor goodness-of-fit.
Therefore, we present an information-reduced approach in the File S2 that increased the signal in $y$ by summation of counts across $J_i$ survey days on which the $i^{th}$ camera was active. If wild boar change their space use throughout the diel cycle (H2), we expect that the highest-ranking model includes a spatio-temporal effect, *i.e.*, $f_3(t, lon(i), lat(i))$. If wild boar avoid human disturbance throughout the day (H3), high trapping rates at daytime should be concentrated in the centre of the forest. For wild boar to have a different space use during the night compared to the day (H4), their spatial pattern during the night should be uncorrelated with that observed during the day. To test this, we first averaged $\widehat{\lambda_{ijt}}$ across all days $J$ and then calculated Pearson correlations $corr(\overline{\lambda_{it1}}, \overline{\lambda_{it2}})$ between all pairwise combinations of solar hours.

*Hunting pressure*
For data on hunting activities which occurred at hunting location (hunting post) $s = 1, 2, \ldots, S$ and at solar hour $t$, we adopted a similar strategy as for wild boar observations: we used exact solar times of hunting attempts $t_h^*$ to obtain and compare activity peaks, as well as overall activity levels using *fitact()* and *compareAct()*. After mapping clock times of observations to solar hours $t$, we used a GAMM to estimate hunter space use across the diel cycle. Specifically, we assumed the number of hunters $h_{st}$ present at hunting post $s$ at solar hour $t$ to follow a negative binomial distribution:

$$h_{st} \sim NegBin(\lambda_{st}^h, \theta^h),$$
$$\text{and,}$$
$$log(\lambda_{st}^h) = log(J) + f_1^h(t) + f_2^h(lon(s), lat(s)).$$

Note that we used the total number of survey days $J$ as an offset term, such that the hunting rate $\lambda_{st}^h$ represents the expected number of hunters at hunting post $s$ during solar hour $t$ of any given day (instead of the expectation across all days). Moreover, for hunting records we did not model the full (3d) tensor product as before, since there was too little data available at many solar hours $t$. Instead, we modelled $f_1^h$ as a separate cubic cyclic regression spline and $f_2^h$ as the superposition of $f_{lon}^h(lon)$ and $f_{lat}^h(lat)$, again with the number of knots $k = 5$ for each of these terms. To test correlations between diel space use of wild boar and hunters, we first averaged $\widehat{\lambda_{ijt}}$ across all days $J$ and then calculated Pearson correlations $corr(\overline{\lambda_{it}}, \widehat{\lambda_{st}^h})$ for solar hours $t$ with at least one hunting record. If hunting pressure influences the diel space use of wild boar (H5), we expect to observe negative correlations between wild boar space use and hunting space use at solar hours where hunting takes place.

## RESULTS

### Wild boar activity and space use
During the growing season, wild boar displayed a bimodal activity pattern across all of the management zones in Meerdaal, with peaks at sunrise ($\pi/2$) and just after sunset ($3\pi/2$) (Fig. 2). Moreover, wild boar activity remained high throughout the night ($3\pi/2 - \pi/2$) compared to the day ($\pi/2 - 3\pi/2$), when there was almost no activity. Timing of the

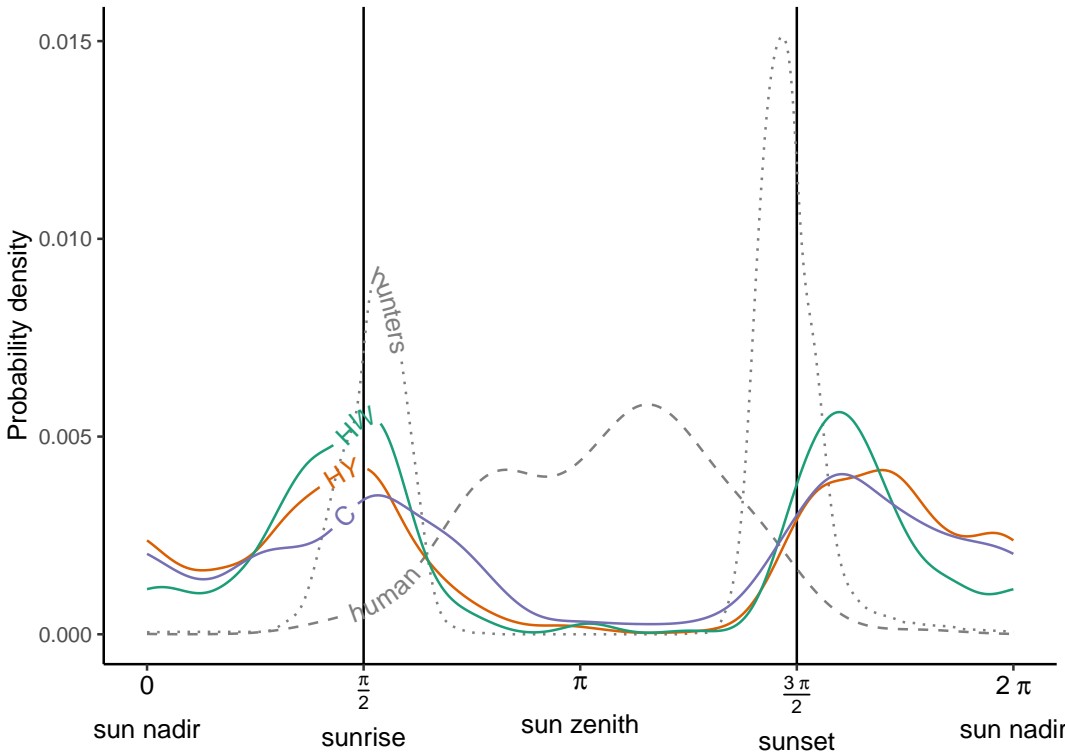

**Figure 2  Activity patterns of humans, hunters and wild boar.** Human and hunter activity density are illustrated by the dashed and dotted curves, respectively. Wild boar activity densities in the year-round hunting zone (HY—red curve), winter hunting zone (HW—green curve) and core zone (C—blue curve). Vertical lines indicate times of sunrise and sunset.

**Table 3  The timing and magnitude of the first and second activity peaks for different populations.**

| Population | 1st peak | | 2nd peak | |
|---|---|---|---|---|
| | Solar time (rad) | Probability density | Solar time (rad) | Probability density |
| Human | 2.479 | 0.004 | 3.645 | 0.006 |
| Hunters | 1.694 | 0.009 | 4.602 | 0.015 |
| Year-round hunting zone | 1.534 | 0.004 | 5.117 | 0.004 |
| Winter hunting zone | 1.620 | 0.005 | 5.081 | 0.005 |
| Core zone | 1.706 | 0.004 | 5.154 | 0.004 |

peaks in the three zones were well aligned both at sunrise ($\pi/2$) and after sunset ($3\pi/2$) (Tables 1 and 3). Additionally, hunting activity peaks coincided with maximum wild boar activity only at sunrise. Human activity largely occurred when boars were inactive (Fig. 2; Table 3). Overall, wild boar were active during 51%, 39% and 49% of the day in the zones HY, HW and C, respectively. Differences between zones in percent of total time active were statistically significant for HY *vs.* HW ($\Delta = 0.13$, $W = 7.82$, $p = 0.005$), buy not for HW *vs.* C ($\Delta = -0.09$, $W = 3.28$, $p = 0.07$) and HY *vs.* C ($\Delta = 0.03$, $W = 0.21$, $p = 0.64$).

According to AIC, a model including a random effect for trapping day, a cyclic smoother for solar time and week of the year, and a 3d smoother for the combination of solar time, longitude and latitude (M5) substantially outperformed all other candidate models, except the full model (M6; $\Delta$AIC = 1.47) (Table 2). The strongest drops in $\Delta$AIC were observed when adding a spatial smoother (from M3 to M4) and a spatiotemporal smoother (from M4 to M5) to the model structure. The QQ plots in Figs. S3.1 and S3.2 suggest that a negative binomial version of M5 fit the wild boar counts better than the Poisson alternative. However, the distribution of the deviance residuals was dominated by small negative values and observed *versus* fitted values resembled a funnel (Fig. S3.1). M5 revealed that wild boar trapping rate randomly varied from day-to-day, with some months having consistently lower or higher encounter rates, *e.g.,* June–July 2020 (Fig. 3A). Moreover, it showed that wild boar trapping rate during the growing season peaked in late June–early July (Fig. 3B). At a daily scale, the trapping rate displayed a bimodal curve with peaks at sunrise and sunset (Fig. 3C, cfr. activity patterns obtained by kernel density estimation in Fig. 2). Projecting the model predictions for mean trapping rates of the highest ranking model (M5) on our study area revealed that wild boar space use during active times (around activity peaks) was mostly restricted to the south of Meerdaal (*i.e.,* lower part of HY), while boar selected for the centre (*i.e.,* HW and C) of the study area during daytime (Fig. 4). The percentages of variance explained in models without a spatial effect (M3), with a spatial effect (M4) and with a spatiotemporal effect (M5) revealed an increasing contribution of the spatial/spatiotemporal smoothers to the total variance (Table 4). Finally, *post hoc* comparisons between projected model predictions revealed that pairwise correlations of wild boar trapping rates were positive and statistically significant for most combinations of solar hours. Only the solar hours around the sun's nadir and zenith were uncorrelated with each other (Fig. 5).

### Hunting pressure—landscape of fear

Similar to wild boar observations, a negative binomial model fitted the hunter data better than a Poisson GAMM (Figs. S3.3–S4). According to the negative binomial GAMM, hunters were predominantly active in the periphery of Meerdaal, except for small regions in the southwest and northeast of the study area (Fig. 4). During times of wild boar activity, positive Pearson correlations between the space use of hunters and boars were significant for solar hours between $(18/24) \, 2 \, \pi$ and $(21/24) \, 2 \, \pi$ (Table 5).

## DISCUSSION

The objective of this study was to estimate diel space use—space use patterns across the diel cycle—of wild boar from camera trapping data in the context of an agro-ecosystem where hunting occurs. For this purpose, we used GAMMs because they allow the construction of a single smoother as a function of a set of coordinates and time of the day (solar time), while at the same time specifying different types of smoothing for each variable (*Pedersen et al., 2019*).

Activity patterns based on circular kernel densities (*activity* package) show that wild boar in Meerdaal are almost exclusively nocturnal across all three management zones.

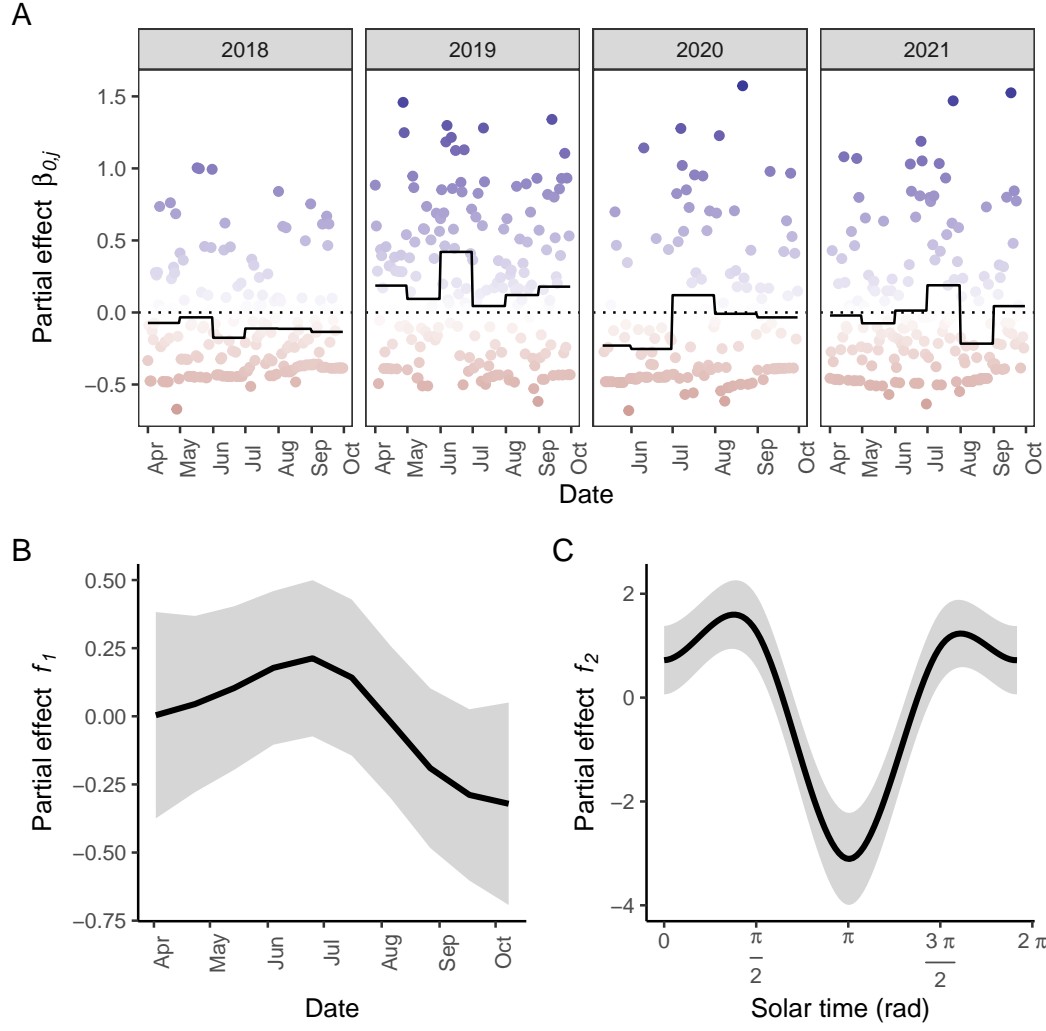

**Figure 3 Partial effects of the elements in M5 on wild boar trapping rate.** (A) Random effects per day (circles; blue: positive effects, red: negative effects) and the corresponding averages per month (full line; the dotted line marks the zero-mean effect size). (B) Effects of the week of the year. (C) Effects of the solar time in radians. Panels B–C: Mean effect size as a function of the date/solar time are indicated by the black lines; grey areas represent 95% confidence intervals. Effects of the tensor product of longitude, latitude and solar time ($f_3$) were excluded for visual clarity.

However, we also observed strong peaks in wild boar activity at sunset and sunrise, typical of crepuscular activity. Hence, our results only partially support nocturnal wild boar activity in Meerdaal (H1). Activities inferred from our GAMM yield similar insights in the activity periods of wild boar. The almost exclusively nocturnal activity that we observed for wild boar is consistent with activity patterns reported in other studies (*Brivio et al., 2017*; *Keuling, Stier & Roth, 2008b*; *Wevers et al., 2020*). The nocturnal activity of wild boar has been linked to an avoidance of human disturbance (*Gaynor et al., 2018*; *Podgórski et al., 2013*). The strong peaks at sunset and sunrise that we observe appear inconsistent with these studies. However, continuous activity of wild boar during short summer nights at high

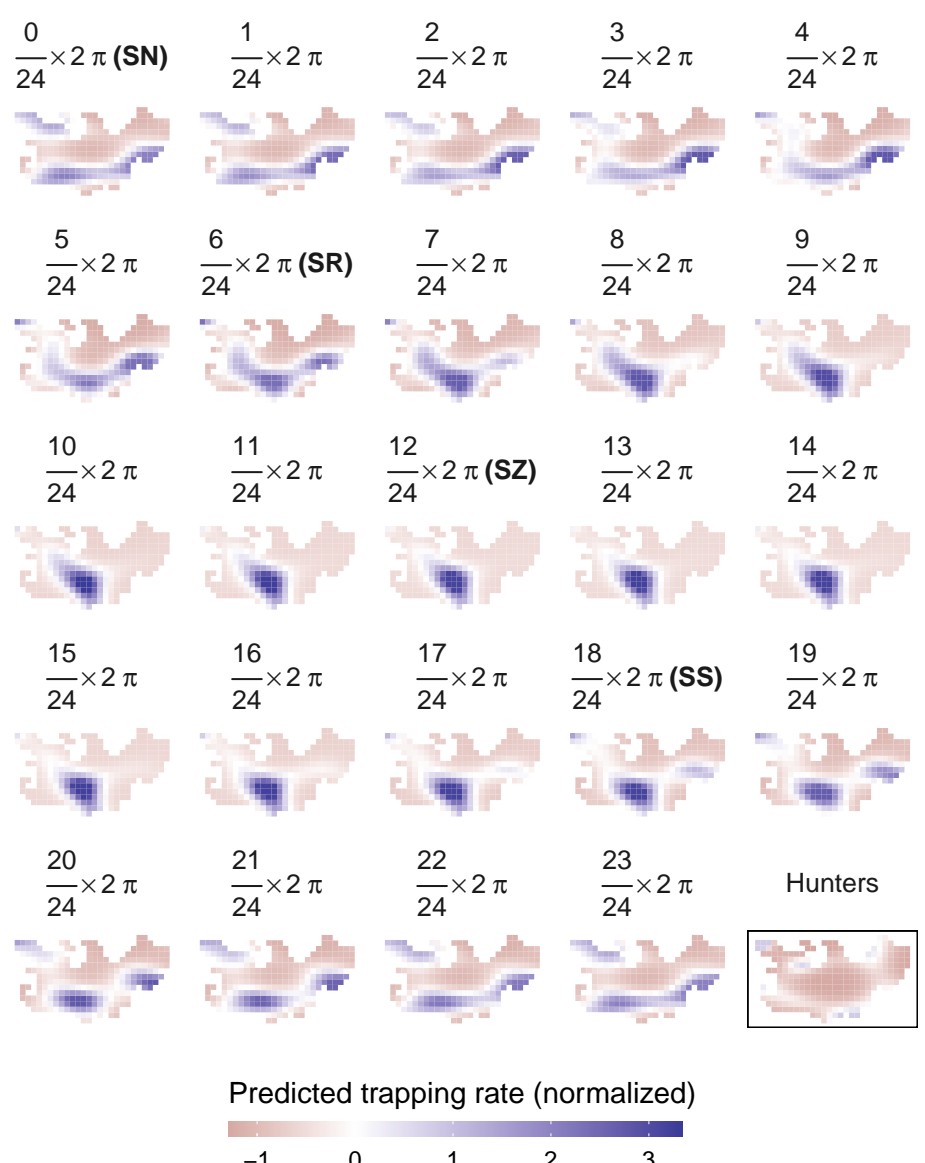

$\frac{0}{24} \times 2\pi$ **(SN)**    $\frac{1}{24} \times 2\pi$    $\frac{2}{24} \times 2\pi$    $\frac{3}{24} \times 2\pi$    $\frac{4}{24} \times 2\pi$

$\frac{5}{24} \times 2\pi$    $\frac{6}{24} \times 2\pi$ **(SR)**    $\frac{7}{24} \times 2\pi$    $\frac{8}{24} \times 2\pi$    $\frac{9}{24} \times 2\pi$

$\frac{10}{24} \times 2\pi$    $\frac{11}{24} \times 2\pi$    $\frac{12}{24} \times 2\pi$ **(SZ)**    $\frac{13}{24} \times 2\pi$    $\frac{14}{24} \times 2\pi$

$\frac{15}{24} \times 2\pi$    $\frac{16}{24} \times 2\pi$    $\frac{17}{24} \times 2\pi$    $\frac{18}{24} \times 2\pi$ **(SS)**    $\frac{19}{24} \times 2\pi$

$\frac{20}{24} \times 2\pi$    $\frac{21}{24} \times 2\pi$    $\frac{22}{24} \times 2\pi$    $\frac{23}{24} \times 2\pi$    Hunters

Predicted trapping rate (normalized)

−1    0    1    2    3

**Figure 4** **Predicted spatiotemporal variation in normalized wild boar trapping rates across 24 solar hours.** Normalized hunting pressure (multiplied by factor 10 for visual clarity) is illustrated by the map inside the rectangle. SN, sun nadir; SR, sunrise; SZ, sun zenith; SS, sunset.

latitudes, which even extend after sunrise or before sunset, have been reported (*Keuling, Stier & Roth, 2008b*). Most likely, nights during the summer are too short for wild boar to meet their energetic requirements. Other studies have even observed a unimodal activity pattern for wild boar, with a peak in activity around midnight (*Caruso et al., 2018*; *Johann et al., 2020*). Several hypotheses could have led to the crepuscular-like activity pattern that we observed for wild boar in our study area. Possibly, lower probability of detection by cameras during the night compared to daylight hours could explain the apparent reduction in activity across the night (*Palencia et al., 2022*). Alternatively, it could be that wild boar

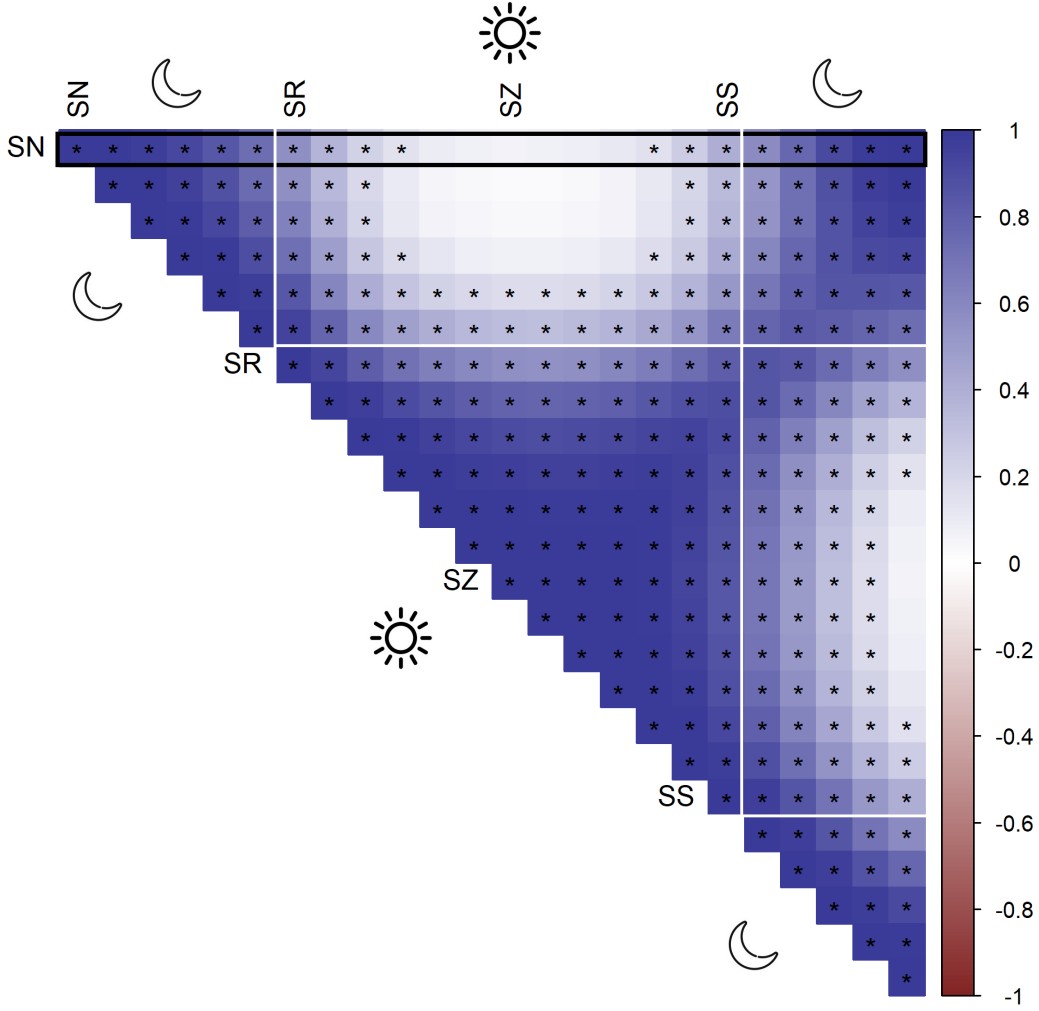

**Figure 5** **Pairwise Pearson correlations between maps of predicted wild boar trapping rates at each solar hour.** Negative correlations (red), positive correlations (blue), significant correlations are marked by asterisks. The black rectangle highlights the pairwise comparisons for sun nadir (SN), including those with sunrise (SR), sun zenith (SZ) and sunset (SS).

stay within the forest during the night (*i.e.,* available for detection), but that they are engaged in comfort-related behavior (*i.e.,* not moving and thus not generating detections) (*Erdtmann & Keuling, 2020*). Another possibility is that wild boar move into adjacent agricultural fields to forage around sunset and return at sunrise. While commutes between the forest and agricultural fields have been observed by *Keuling, Stier & Roth (2009)*, they did not report commutes of a daily frequency. If wild boar commute on a daily basis in our study area, this could lead to more detections clustered at sunset/sunrise. At the same time, this would lead to fewer nighttime detections in the forest (*i.e.,* the area monitored in our study), simply because wild boar are temporarily unavailable in this area. Many wild boar, especially female yearlings may even shift their home range permanently to agricultural fields during the summer (*Keuling, Stier & Roth, 2008a*; *Keuling, Stier & Roth,*

**Table 4  Variance components of non-spatial, spatial and spatiotemporal models.** Standard deviation SD and its 95% confidence interval, variance and percentage of variance explained by the partial effects of models M3 (non-spatial), M4 (spatial) and M5 (spatiotemporal) are presented.

| Model | Effect | SD | 0.025 | 0.975 | Variance | % Variance explained |
|---|---|---|---|---|---|---|
| No spatial effect | $\beta_{0,j}$ | 0.835 | 0.242 | 2.881 | 0.697 | 3.425 |
| | $f_1(t)$ | 4.433 | 1.284 | 15.300 | 19.653 | 96.573 |
| | $f_2(week(j))$ | 0.019 | 0.009 | 0.043 | 0.000 | 0.002 |
| Fixed spatial effect | $\beta_{0,j}$ | 0.820 | 0.184 | 3.645 | 0.672 | 3.240 |
| | $f_1(t)$ | 4.410 | 1.991 | 9.767 | 19.447 | 93.710 |
| | $f_2(week(j))$ | 0.027 | 0.012 | 0.060 | 0.001 | 0.003 |
| | $f_3(lon(i), lat(i))1$ | 0.708 | 0.343 | 1.461 | 0.502 | 2.417 |
| | $f_3(lon(i), lat(i))2$ | 0.361 | 0.081 | 1.607 | 0.131 | 0.630 |
| Spatiotemporal effect | $\beta_{0,j}$ | 0.776 | 0.127 | 4.751 | 0.603 | 1.765 |
| | $f_1(t)$ | 5.247 | 2.668 | 10.319 | 27.536 | 80.652 |
| | $f_2(week(j))$ | 0.027 | 0.012 | 0.063 | 0.001 | 0.002 |
| | $f_3(t, lon(i), lat(i))1$ | 2.202 | 1.251 | 3.875 | 4.849 | 14.201 |
| | $f_3(t, lon(i), lat(i))2$ | 0.832 | 0.525 | 1.318 | 0.693 | 2.029 |
| | $f_3(t, lon(i), lat(i))3$ | 0.679 | 0.111 | 4.156 | 0.461 | 1.351 |

**Table 5  Pearson correlations $\rho$ and their significance between maps of wild boar diel space use and hunting pressure for solar hours with >1 hunting record.**

| Solar hour | Statistic | $\rho$ | p-value | Significance |
|---|---|---|---|---|
| $(3/24)\cdot 2\pi$ | 1.235 | 0.0760 | 0.218 | ns |
| $(4/24)\cdot 2\pi$ | 1.316 | 0.0810 | 0.189 | ns |
| $(5/24)\cdot 2\pi$ | 1.447 | 0.0890 | 0.149 | ns |
| $(6/24)\cdot 2\pi$ | 1.548 | 0.0950 | 0.123 | ns |
| $(7/24)\cdot 2\pi$ | 1.375 | 0.0840 | 0.170 | ns |
| $(8/24)\cdot 2\pi$ | 0.973 | 0.0600 | 0.331 | ns |
| $(15/24)\cdot 2\pi$ | 0.662 | 0.0410 | 0.509 | ns |
| $(16/24)\cdot 2\pi$ | 1.090 | 0.0670 | 0.277 | ns |
| $(17/24)\cdot 2\pi$ | 1.611 | 0.0980 | 0.108 | ns |
| $(18/24)\cdot 2\pi$ | 2.074 | 0.1300 | 0.039 | * |
| $(19/24)\cdot 2\pi$ | 2.325 | 0.1400 | 0.021 | * |
| $(20/24)\cdot 2\pi$ | 2.370 | 0.1400 | 0.019 | * |
| $(21/24)\cdot 2\pi$ | 2.203 | 0.1300 | 0.029 | * |
| $(22/24)\cdot 2\pi$ | 1.853 | 0.1100 | 0.065 | ns |
| $(23/24)\cdot 2\pi$ | 1.494 | 0.0910 | 0.136 | ns |

**Notes.**

*p-value $\leq$ 0.05 (ns); 0.05 $\geq$ p-value > 0.01 (*); 0.01 $\geq$ p-value $\geq$ 0.001 (**); and p-value $\leq$ 0.001 (***).

*2009*). Typically, these individuals also display increased diurnal activity (*Keuling, Stier & Roth, 2008b*). In our study area, we observed very few wild boar during daytime, which could result from the absence of activity data from agricultural fields adjacent to Meerdaal.

To further investigate the spatiotemporal patterns in wild boar trapping rate across the diel cycle, we fitted a selection of GAMMs. From the highest ranking GAMM, it appears

that there were more days with low wild boar trapping rate during the period of June–July 2020 as compared to other months in the study period. This period was right after the most stringent COVID-19 related lockdown in Belgium, in which all non-essential travel was prohibited (data from this lockdown period was excluded from the analysis). Both positive and negative impacts of COVID-19 related suppression of human activity on the detectability of a species have been observed (*Anderson, Waller & Thornton, 2023*; *Nicosia et al., 2023*; *Procko et al., 2022*). In our study area, human activity, especially hiking, increased during the stringent lockdown of April–May 2020. This may have led to reduced activity and thus the lower number of wild boar detected during and right after the lockdown period. Regardless of the year, we also found that wild boar trapping rate peaked at the beginning of July. This is consistent with the increased wild boar activity during the summer observed in other studies (*Brivio et al., 2017*; *Johann et al., 2020*). Increased trapping rates around July could be a consequence of cereals, such as wheat, being ripe at that time resulting in more commutes between the forest and surrounding agricultural fields in Meerdaal (*Keuling, Stier & Roth, 2008b*; *Keuling, Stier & Roth, 2009*; *Kramer et al., 2022*). In addition, females, which typically have high energetic requirements in the summer in order to nurse their piglets (until they are about 4 months old), may also contribute to more detections during this period of the year (*Keuling, Stier & Roth, 2008b*). The highest ranking GAMM also included a spatiotemporal effect, which supports the hypothesis that wild boar in our study area do change their diel space use (H2). ΔAICs were particularly large between non-spatial and spatial models, and spatial and spatio-temporal models. Moreover, an increasing percentage of the total variance explained was attributed to spatial/spatiotemporal effects. Together, these findings reinforce that wild boar space use is not fixed throughout the diel cycle. Our study also appears to support the hypothesis that wild boar stay in centre of the forest during the day (H3), but that they utilized a larger area, including sites near the forest edge, at nighttime (H4). However, this support relied solely on visual inspection of the spatial patterns. These patterns revealed a concentration of high trapping rates in the centre during the day, but not during the night. Therefore, it is uncertain which factors are the true drivers of the spatiotemporal variation, observed in our study. Presumably, wild boar select for the centre of the forest to avoid human disturbance when resting (*Bollen et al., 2024*) and areas near the forest edge to be close to foraging grounds (*i.e.,* agricultural fields) (*Bollen et al., 2024*; *Keuling, Stier & Roth, 2008a*; *Thurfjell et al., 2009*). However, diurnal activity of wild boar may also be concentrated to the centre of the forest (HW and C) because the hunting pressure in the central zones is lower (*Johann et al., 2020*). A possible avoidance for this zone at times of human activity could be exacerbated by the combination of lethal (*i.e.,* hunting) and non-lethal (*e.g.,* hiking) human activities (*Paton et al., 2017*).

In our study area, wild boar did not seem to temporally avoid hunters when active, as observed elsewhere (*Johann et al., 2020*; *Ohashi et al., 2013*). However, the absence of a statistical effect of hunting does not necessarily mean that a biological effect is not present. Moreover, we warn that the results of our study systems may not apply to other studies. For instance, the hunting pressure in Meerdaal, as compared to other study areas, may be too low for wild boar to shift their activity patterns. Alternatively, it could be that wild
boar do not temporally avoid hunters during our study period because the short summer nights are too short for them to meet their energetic requirements (*Keuling, Stier & Roth, 2008b*). In that case, wild boar may still avoid hunters spatiotemporally, which we assessed using a GAMM based on records of hunters. The landscape of fear that we infer from this GAMM was significantly (positively) correlated with wild boar diel space use around sunset, starting from $(18/24)\ 2\ \pi$ through $(21/24)\ 2\ \pi$. Moreover, adding the effect of hunting to the GAMM modelling diel space use of wild boar did not yield a better model according to AIC. We also found that the effect of hunting was not significant. This suggests that our last hypothesis (H6) should be rejected, such that wild boar do not avoid hunters spatiotemporally in Meerdaal. However, we consider it likely that hunters in our study area preferentially visit locations of high wild boar trapping rates at times when wild boar are active, which has been proposed by others (*Wevers et al., 2020*). Provided that hunters select the same areas that are intensely used by wild boar, the latter may also trade off their need for food intake with the risks induced by the hunters (*Ferrari, Sih & Chivers, 2009*). This is in accordance with some other studies, which found that wild boar space use is primarily driven by food resources and that they are seemingly insensitive to predation risk (*Bubnicki et al., 2019*; *Wevers et al., 2020*). Furthermore, wild boar in Meerdaal may trade off avoidance of non-lethal human activity with the risks induced by hunters (*Bollen et al., 2024*). Thus, our data suggest that there is no substantial impact of hunting on diel space use. However, we cannot completely rule out an effect of hunting because we lack information on how hunting may have impacted the space use of wild boar in agricultural fields adjacent to our study area.

We did not obtain samples from the agricultural fields adjacent to Meerdaal, is arguably the most important limitation of our study for two reasons. First, it prevents us from assessing the full impacts of hunting. Second, we observed wild boar gradually moving toward the forest edge during the night but lack information on the situation beyond the forest edge. Wild boar are known to either use agricultural fields temporally or permanently during the summer or even year-round (*Amici et al., 2012*; *Keuling, Stier & Roth, 2009*; *Thurfjell et al., 2009*). Thus, the diel space use patterns inferred from our camera trapping network are likely to shed an incomplete light on their space use patterns within the broader region around the forested area in Meerdaal. The need for relatively large sample sizes, given that few photo-captures will typically be produced during times of inactivity, is another limitation of our approach. This may make our approach unsuitable for short-term camera trapping studies and for rare or inconspicuous species. In order to produce reliable diel space use maps, we had access to observations from 9,542 trapping days for all cameras combined. Even with this large number of data points, the errors associated with spatiotemporal predictions of diel space use are substantial. Furthermore, our GAMM had problems predicting the rare encounters of a large number of individuals that occur from time to time, since most solar hours had a zero-count (99.52%). This behavior was reflected in the residual plots. One solution is to fit a GAMM to counts aggregated over all survey days, hence only retaining information on the solar hours and spatial locations. This lowered the percentage of solar hours having a zero-count considerably (72.00%) at the cost of losing information about calendar dates of the observations. Nevertheless, we

found that this strategy preserved the typical diel space use of wild boar in our study area. So when the only goal is to obtain diel space use, without acknowledging other sources of variation (between days, weeks, months or years), this reduced information approach can be adopted.

Another drawback of our GAMM is that it does not account for false negatives (*i.e.*, imperfect detections) as is done in other popular modelling frameworks (*Dénes, Silveira & Beissinger, 2015*; *Guillera-Arroita, 2017*). The failure to correct for imperfect detections may possibly introduce bias in the space use patterns inferred from a GAMM. In principle, occupancy (*MacKenzie et al., 2002*) or N-mixture (*Royle, 2004*) type of models, which account for imperfect detections, can be used to model nearly continuous changes throughout the diel cycle. However, as these models require repeated samples in space and time to estimate occupancy/abundance for each solar hour, seasons would need to be of 1-h length and surveys of <1-h. Not only would this model not make sense biologically, it would also be computationally infeasible to fit. Finally, landscape of fear maps that we inferred from hunting pressure could have been distorted by non-random missing data or underreporting of hunter visits (∼10–20% of hunting records was missing/not reported).

## CONCLUSIONS

The main objective of our study was to infer the diel space use patterns of wild boar in an agro-ecosystem, where hunting occurs, from camera trap data. We revealed that wild boar in Meerdaal were mostly nocturnal (H1), with additional crepuscular activity. Moreover, we found that wild boar in our study area adjusted their space use pattern throughout the diel cycle during the growing season (H2). This was possibly to avoid human activities during daytime, as indicated by a selection for the centre of the forest (H3). We also found that wild boar space use during the night, when they utilized areas in the periphery of Meerdaal, was uncorrelated with its space use during the day (H4). In the future, placing cameras in the agricultural fields adjacent to the study area could help to provide information on the strength of the attraction to agricultural fields when crops are growing. Finally, we did not find sufficient evidence to support our hypothesis that wild boar in Meerdaal spatiotemporally avoided hunters (H5), which does not mean that a biological effect of hunting was absent.

More generally, we have shown that GAMMs, despite some limitations, can be useful tools to model diel space use from photo-captures. However, to test the robustness of camera traps, which do not record individuals when they are inactive, for inference on diel space use, we urge that our approach be compared to telemetry-based methods. Moreover, future studies could improve our approach in several ways. First, some of the GAMMs for the modelling of diel space use had relatively poor goodness-of-fit. The application of a piecewise exponential additive model (a GAM(M) for exponentially-distributed responses) to time-to-event data may partially resolve this in the future (*Bender, Groll & Scheipl, 2018*). Essentially, this would be an extension of the time-to-event model in *Moeller, Lukacs & Horne (2018)* that permits the modelling of smooth predictor-response relationships. We also encourage the extension of the detection function in occupancy models that

simultaneously estimate diel activity and occupancy, for instance the model in *Kellner et al. (2022)*, to incorporate spatial changes in diel activity through 24-h periods (*i.e.,* diel space use). Note that this would still only produce a single occupancy map across the day. However, the detection probability would vary in space–time similar to trapping rates inferred from the GAMMs that we presented. Hence, if the prime interest is in modelling availability/trapping rate, we suggest applying the much simpler methods presented in this article. If the overarching occupancy pattern is of importance, we suggest that researchers implement our approaches into the detection function of occupancy models. Another interesting development in the modelling of diel space use could be the implementation of Gaussian processes, a parametric alternative to spline approaches (*Williams, 2006*). Finally, treating hunter counts and wild boar counts as two correlated processes, analyzed through a joint modelling approach for preferentially sampled data, may improve inference on hunting effects (*Diggle, Menezes & Su, 2010*).

## ACKNOWLEDGEMENTS

We are grateful to the Flemish Agency for Nature and Forest and the local nature conservation NGO ''Vrienden van Heverleebos en Meerdaalwoud'' to allow us to place camera traps on their properties. Further, we thank all volunteers and students that aided in the field or processed and annotated photographs. Our final word of gratitude goes to Donald Kramer, Oliver Keuling and Frederik Dalerum for providing us with valuable feedback, which has improved both the form and content of this article.

### Funding

This work makes use of data and/or infrastructure provided by INBO and funded by Research Foundation-Flanders (FWO) as part of the Belgian contribution to LifeWatch. Martijn Bollen is a PhD fellow funded by a BOF mandate at Hasselt University. Thomas Neyens received funding from the FWO (G0A4121N) and from the Internal Funds KU Leuven (project number 3M190682). The funders had no role in study design, data collection and analysis, decision to publish, or preparation of the manuscript.

### Grant Disclosures

The following grant information was disclosed by the authors:
Research Foundation-Flanders (FWO) as part of the Belgian contribution to LifeWatch.
BOF mandate at Hasselt University.
the FWO (G0A4121N) and from the Internal Funds KU Leuven:  3M190682.

### Competing Interests

The authors declare there are no competing interests.

## Author Contributions

- Martijn Bollen conceived and designed the experiments, analyzed the data, prepared figures and/or tables, authored or reviewed drafts of the article, and approved the final draft.
- Jim Casaer conceived and designed the experiments, performed the experiments, analyzed the data, authored or reviewed drafts of the article, and approved the final draft.
- Thomas Neyens analyzed the data, authored or reviewed drafts of the article, validation of the data analysis, and approved the final draft.
- Natalie Beenaerts conceived and designed the experiments, authored or reviewed drafts of the article, supervision, and approved the final draft.

## Data Availability

The pre-processed wild boar activity and hunter activity data are available in the Supplementary Files.

## Supplemental Information

Supplemental information for this article can be found online at http://dx.doi.org/10.7717/peerj.17390#supplemental-information.

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
