# Peer review of "When and where? Day-night alterations in wild boar space use captured by a generalized additive mixed model"

_PeerJ, doi:10.7717/peerj.17390_

## Round 0.1 · original submission · Major Revisions

Overview

This study used camera trap data to investigate daily activity patterns of wild boar, hunters, and other humans from spring through fall during four years in a Belgian nature reserve that was divided into three zones with different levels of hunting pressure. The boars forage in adjacent agricultural fields at night, and the specific goal was to see whether this was reflected in more activity in the central areas during the day and in peripheral areas during the night and whether differences in hunting pressure resulted in changes in activity pattern. The authors first examined mean activity patterns, as measured by the number of camera events, over all camera locations, days, and years, using kernel density functions for boars in each zone, hunters (overall) and, presumably, non-hunting humans (the latter shown in a figure and mentioned in text, but probably not in Methods). This showed strong peaks in boar activity at sunrise and just after sunset. Hunter activity also showed two strong peaks, coinciding with the peak in boar activity at sunrise but preceding it at sunset. Other human activity occurred throughout the day and peaked in mid-afternoon. Total activity was similar in the zone allowing year-round hunting and in the zone with almost no hunting but less in the zone that allowed hunting only during the winter. To examine the correlates of the spatial and temporal patterns, the authors used GAMs, including time of day, day, week, longitude and latitude as possible independent variables for the occurrence of boars and hunters. The best-fitting GAM included an effect of day (which I don’t think I understand), an effect of week showing changes in encounters through the season with a peak in late June – early July, an effect of time of day showing bimodal activity as with the previous analysis, and an effect of location indicating diel changes in space use (which I found very challenging to grasp from the figure). Including hunting effort was stated not to improve the model for boar activity, but I did not find a clear explanation in the Methods. Spatial variation in hunting effort was described in the text, but I had trouble verifying it from the figure. There was a positive correlation between the numbers of hunters and boars for three hours preceding sunrise but not before or after sunset. The authors concluded that boars adjust their space use to avoid human activity during the day and to use adjacent agricultural areas at night and that GAMs are a useful tool to investigate spatiotemporal patterns.

This is a challenging study because its focus is on changes in space use over the diel cycle but it does not have data on the main alternative area (adjacent agricultural fields) or on the degree of reduced detectability at night. The effect of hunting is hard to assess because of the distribution of hunting zones and lack of any zone with no hunting but two zones with no hunting during the study period. Furthermore, it includes data from one year of Covid lockdown during which space use by the boars may have greatly changed. Both reviewers indicate that the study is potentially publishable but requires substantial revision. I find the manuscript needs a clarification of the objectives in light of the opportunities and limitations of the data set, more rigorous examination of the evidence that space use varies over time and that this is due to avoidance of humans. I was unable to find clear evidence for space use variation within the reserve over time except for visual inspection of 24 tiny graphs in Fig. 2, 3 and a supplemental figure. The Discussion needs to be more rigorous and better organized in relation to the findings and the objectives of the study. The figures need substantial improvement to convey the findings effectively.

The use of English is generally quite strong, except for the awkward use of the present tense as noted by both reviewers. The authors should use the past tense to describe methods and results, present tense to describe broader generalizations. The paragraph structure is not always clear; it would be helpful to either skip an extra line between paragraphs or indent the first sentence.

You can consider my comments below as if they are a third review: make appropriate changes if they are valid or explain clearly and in detail if they are not valid. I am broadly familiar with habitat selection but lack expertise in some of the statistical approaches to it. However, I do not feel that I am more limited than many potential readers would be. In addition to the comments below, I used highlights with inserted comments directly on the pdf to indicate potential problems and suggest replacement wording. This included highlighting the inappropriate use of present tense. When there was a repeated occurrence, I sometimes used only highlights without inserted comments. You do not have to respond to all these minor suggestions in your rebuttal unless you disagree and have not made the suggested changes. I may have missed some repeated errors, so you should check the entire text for similar errors (including any new text, of course).

Editor’s Comments
Abstract
• Remove the first paragraph of the Abstract. It is too long and includes too much background.
• L19 and elsewhere. When a number is used as an adjective, a hyphen between number and units is required: ‘24-hour cycles’. I am not certain but presume that this rule also applies for abbreviations: ‘24-h cycles’
• L25. Give the scientific name as first mention of the study species.
• Why is there no mention of the activity pattern result or of the hunter pattern and correlation with boars?

Introduction
• L49-50. This is an odd collection of references (one fish study, one on voles, one on zebras) to support the widespread occurrence of diel changes in space use. Can you replace with one or two appropriate reviews?
• L86-87. Not clear why the boar should be in the centre of the forest rather than closer to the fields; not clear why they should be in the forest close the fields – is there foraging available there?
• L87. There should be a hypothesis regarding the effect of hunting.
• Because you argue that GAMs are a useful tool for investigating diel space use patterns, you need a paragraph indicating techniques that are used for this approach and what their advantages and limitations are.
• Restriction of the study to the growing season and the reason for this should be stated and justified in the Introduction.

Methods
• L95. Fig. 1 should be only a map of the reserve, not the activity pattern which is part of the results.
• Fig. 1. I did not find the map very helpful. Perhaps more detail, including latitude and longitude marks on the frame, perhaps an inset showing the location within Belgium/Europe, perhaps more detail of vegetation types, roads, and trails, and the boundaries of the different zones, and perhaps extending the map to show the location of relevant agricultural fields. Note that using color-coding exclusively prevents comprehension by readers who are red-green color blind or who are reading a black-and-white printed version.
• In the caption to Fig. 1, explain the multiple overlapping crosses. If this is to indicate repeated use of the same location, is it really needed on this map?
• L97. Are there relevant seasonal changes in food availability within the reserve, for example acorns in late summer?
• L105. ‘from November until March’ could imply that November and March are not included. If November and March are included, you could write ‘restricted to November through March’.
• There is no hunting during the study period in two zones, but this is not clear because you have not indicated the study period yet. This would be the place to emphasize the spatial pattern of combined no-hunting zones during the study period.
• L102. Do the areas differ besides in hunting pattern, for example in vegetation or in other ways that would make them different in addition to hunting pressure? How are areas differentiated on the ground?
• L106 Not clear what you mean by joint hunting. Is this a group hunt?
• L108. Can boars learn where the hunting stands are and where the danger zones around them are?
• L110. Not clear why you say night hunting is ‘also’ allowed, since the given times cover the night
• L114-115. I don’t think you need this subhead. You can replace ‘Wild boar’ with ‘Wild boar activity’.
• L116ff. Camera make and model?
• How effective are the cameras at night? Implications for the analysis?
• What is the area in which animals could trigger the camera?
• Are cameras located near stands?
• How much data was lost due to camera defects, accidents, stolen cameras, etc. and how did you treat missing data?
• L127. What is the justification for using a 2-minute interval? This seems very short to be considered separate groups (concern also raised by reviewer).
• L128. Define counts per day for each hour here, not in Table 1. Was any calculation changed if some cameras were not operative on some days?
• L129. This is very late in the Methods to indicate that your study involves only the growing season. The text until now presents information regarding year round effects, so this is a surprise. It should be stated and justified in the Introduction.
• L130 and elsewhere, use a comma to define thousands 10,086.
• L132. I suggest range as well as mean for group size.
• L134. This subhead would be clear if changed to ‘Hunting effort’ or ‘Hunting pressure’.
• L140. Somewhere in this area you need to define hunter visits per day for each hour, not in the caption to Table 1. Also note that Table 1 refers to visits but they do not seem to be defined here. Are they the 10 min periods or the number of hunter records? Does the number of hunters matter?
• L148ff. I am not familiar with ‘solar time’ analysis and its implications. Given the relative recency of the Nouvellet et al (2012) article, other readers are also likely to be unfamiliar with this approach and its implications. Citing Nouvellet et al. is not adequate because this is a semi-theoretical approach dealing with only sunrise. You need to explain how you could anchor your clock times to both sunrise and sunset. This approach seems to imply that the daytime ‘hours’ would be longer than the nighttime hours during the summer and that the ‘hours’ change length throughout the period? This would affect calculations of camera events per hour. I suggest providing a concise description and implications of this approach for interpretation as well as citing a clear article fully laying out the methodology involved.
• L174. Here, lambda is capture rate (what are the units?) On L187, you refer to encounters; is this the same thing? Define the terms, provide units, and use consistently for ease of reader understanding. Note that some readers might consider encounter rate to refer to encounters between boars and hunters because this is how the term is used in predation studies. The term is ok, as long as it is clearly defined.
• L258. What are the units of activity level and what is the possible range?
• L166. Was there a method for identifying the peaks or was this simply by visual inspection?
• L204. It is very important to include somewhere in this section a predictive statement that prepares readers to understand your results in the light of your methods and the limitations of the study. For example, ‘If there is a change in space use over the daily cycle, we would expect solar time, latitude and longitude to be included in the highest ranking model’. This could be followed by how you would determine where the change occurs, for example by plotting the hourly distribution on a map of the area and looking for changes in the location of highest activity. The fact that your Methods restricts the description to the highest ranking model seems to be including Results and is somewhat awkward, according to the normal allocation of material to a research article.
• L224. Again, specify the expectation from your hypothesis: ‘If hunting intensity influences boar distribution, we expect . . .’
• I did not see a method for human recreational activity.

Results
• L229. The activity pattern should be Fig. 2, separate from the study area figure.
• L332. I don’t think you gave a method for documenting non-hunting human activity and producing an activity pattern. Is ‘recreational’ the correct term? Hunting may be considered recreational.
• L233. Rowcliffe et al. refer to the proportion of time active. I think it might be clearer if you referred to this measure as ‘percent of total time active’.
• L233. According to this statement, HW is about 25% lower than HY and C. Recognizing that it is challenging to estimate total under a curve by eye, I think this should be checked. In the figure, the green line which the key indicates as HW is well above the other two lines during the sunrise period and after sunset and does not seem to fall that much below them at other times.
• Fig. 1 (upper panel, to become Fig. 2). This figure and the caption need some work. What are the units of density? I presume that since this figure represents activity, this does not refer to numbers per area. Perhaps it is a probability density, but the peak at greater than 1.0 seems to rule that out. There should be adequate explanation in the Methods text and sufficient clarity in the figure for a reader who looks only at the figure. The x-axis needs more explanation too. It might be useful to add text to the axis label, for example: sun nadir (0), sunrise (π/2), sun zenith (π) etc. Because you use radians in Table 2, you may want a scale in radians also.
• Table 1. Describe what is a count and a visit much more clearly. The calculation per day for each belongs in the methods.
• Table 2. The peaks here are defined in radians but these units are not shown in Fig 1, upper panel. This makes it awkward for readers to compare the figure and table.
• L239. I think it would be more usual to present the AIC results here along with degree of support for the best-fitting model and the occurrence of the different independent variables.
• L240. The order of variables should correspond either to the order used in the Methods or to the magnitude of their effect. The day intercepts are first in the equation but late in the description. Make sure this will all be clear to readers.
• L240. Most biological data show variation, so it is not clear what you mean. Is the magnitude of variation something that is of interest? It is not obvious from examination of the figure that April-June has higher daily variation except in 2019 (positive) and 2020 (negative). Is a bit more description relevant here? Have you tested to be sure that the daily variation is random as stated?
• L245. Since the focus of your entire article is on spatiotemporal variation, I am surprised that you do not have a figure much more informative than Fig. 2D and the brief text. Why not walk the reader through the specific evidence, including mention of the strength of the effect? Understanding partial effects is not easy. My reading of the figure, difficult as it is, does not agree with this interpretation. I see a red (positive) focus near the north most of the time, especially 1.4 to 4.6 which should be sunrise to sunset if I understand correctly. I see a blue (negative) focus extending from the center to the lower edge during the middle part of this. I don’t see any clear center-edge distinction in the first few and last few panels (night).
• L245. Is there importance to the magnitude of the different partial effects?
• L246. It would be more reader-friendly to clarify the difference between 2D and 3 as you develop your evidence. Looking at Fig. 3 and Supp. Fig. 4, I can’t readily see how they differ, except that Fig. 3 includes negative encounters (red), whatever that means. To my eye, hunter encounters are all less than zero. I don’t see any focussed use of the central area during the day.
• Fig. 2. Caption is incomplete. Panels should be indicated by capital letters (see instructions to authors). Panel A. What about dotted line? Referring to the circles as dots is confusing as there is also a dotted line that is not mentioned. Specify time period and years. The caption states that positive partial effects are negative are blue, but the figure shows the opposite. A color blind reader or one reading a photocopy would not be able to use the color information. Is this important? The panels are too close together so that Apr and Oct are jammed together without a space. B. Caption should specify significance of solid line and grey zone. Is there a good reason to present the daily effect by month and the weekly effect by week in two adjacent panels? C. Specify solid line and grey area. D. This is very confusing. Do ‘facets’ refer to panels? Axis labels of latitude and longitude imply that the panels differ in space. Time of day is in decimal radians here but fractions of pi in C. If these are supposed to be hours of the day, it is not clear why there are only 16 panels. Panels are so small as to be nearly impossible to read. What are the black lines in the panels? What is the scale of partial effect for the colors? Are you sure you have red and blue in the correct order here? How would a color blind person be able to interpret the panels? What area is shown in the rectangle? Since the reserve boundary is not perfectly rectangular, either some of the reserve is missing or the (presumed) isolines include non-reserve.
• L254. I cannot recognize the relative higher use of the peripheral areas from the hunter panel in Fig. 3.
• Table 3. Heading is incomplete. It should specify each column head. What is the ‘statistic’ in addition to correlation coefficient and p-value? Should you indicate sample size?
• Modify the supplemental figures and tables in the light of problems raised with the ones in the main text.

Discussion
I am not providing detailed comments on this section because it needs very substantial revision, although I have left in highlights and comments inserted as I read the manuscript. The Discussion should provide inferences and context for the results. However, this Discussion raises a number of empirical findings (for example low encounters in April-June 2020, activity ‘almost exclusively nocturnal’) that should have been clearly documented, preferably quantitatively, first in the results. I also do not find the Discussion sufficiently critical in presenting the strength of the evidence in the light of both the statistical analyses and the constraints of the study system. Finally, different topics are sometimes combined within the same paragraph in confusing ways. A good Discussion should present the topics in similar order to the Results. It should suggest the interpretation of the findings and rigorously address the strengths and weaknesses of the evidence. After this, the findings and interpretations should be related to the broader literature, pointing out similarities and differences. Just as in the Introduction, I think you need a separate paragraph to point out how the GAM helped you understand this situation and how this differed from other possible approaches, because this is a point you make from the title on, but do not fully discuss. It is usually not necessary to cite figures in the Discussion.

Conclusions
PeerJ Instructions to authors asks that Conclusions not be a repetition of the Abstract but focus on broader issues such as future research needs and applications. Some appropriate material for Conclusions is found in the final paragraph of the present Discussion.

References
This section is relatively free of errors, compared to many I see. Thank you. However, there are many cases where italics were omitted from species names and at least one case where capital letters were retained in an article title. Please check them all, not just the ones I highlighted.

Reviewer 1 ·

Basic reporting

Dear authors,
all in all the MS is well written and timely.
I have some general concerns about the length and simplicity.
Not everybody does understand these very mathematical/statistical wordings. The text is quite long and should be shortened to short precise writing also "simple" scientists do understand! eg. Line 107ff is a complicated sentence, please rephrase more simple.
L86: I do not agree with that expectation. Wild boar do prefer sheltered habitats close to food during day. I would not expect them to stay in the center but in some bushes!
follwing: I recomment to formulate some clearly described hypotheses instead of generally formulated thoughts!
All over the text you switch between present and past. Please stay in past for everything which you did. (e.g. line 165, 166, 168 => performed, identified, adopted). Only general findings may be written in present!

Experimental design

Although you dicribe the experimental design, conduction and statistical analyses very elaborately you missed to name the brand and model of the cameras (Line 113)!
I also recomment to show the differnt zones (hunting regimes) in the map.
Line 105 Nov-March (included or excluded?), please be precise!
I would not call "C" a "hunting-free zone, as there are still about 6 hunting events per year. C is "just the zone with the lowest hunting pressure, but not free of hunting (which micht also influence your findings and thus, should also be discussed!
How much hunting is conducted in "HW"? how big is the difference (from a wild boars view)? Think about a landscape of fear!
L151: I doubt daring that the solar time is really the best for these analyses. It is a very good attempt but if you compare soem activity cycles (e.g. in Keuling et al 2008) you might see that wild boar change start of activity in the evening but hardly end of activity in the morning! Another point which could be discussed!

Validity of the findings

no comment

Additional comments

some specific comments:
Activity lowest at noon in forest! But within fields it might be higher in summer (compare e.g. Keuling et al. 2008)
L270ff: For CoVid there is also described here: DOI: 10.1007/s10344-023-01725-8, DOI: 10.1145/3486637.3489494, doi:10.3390/su14084849 , https://doi.org/10.1016/j.gecco.2021.e01895
L283: nursing of piglets stops at an age of 4 months!
L292: in conflict? You may not say so, as there were no CT outside the forest within fields! You miss the activities and movements in other habitats (which are reported by telemetry)
L303: the return at sunlight is not reported this way in Keuling et al. 2009! Wild boar may stay complete weeks in fields, only some may return in the second half of the night!
L305: tomporarily OR PERMANENTLY... !
L318 "In our study area..." This is an important point! Your findings may notr be generalised as long as you cannot compare it to other areas! This point should be highlighted more!
L320 "spatiotemporally in sensu Kohl..." What does that mean, here the simple citation is not enough as not every reader is able to look this up immediately. Needs to be explained in a quick sentence!
L322: It is hard to understand which time this means, better say it more general (half time between midnight and dawn as well as sunrise?)
L366: This is a big backdraw! Please discuss in more detail or otherwise submitt the MS only after this backdraw has been omitted and wild boar are also obesrevd within fields (I know this is a hard job)!

Tab2: In Methods you only describe activity, here you use the term density! How did you calculate densities? => needs to be described in methods. Or is it a mistake and this is a activity-density? => replace term by any better word!
Fig1: Again "density" use correct term (as these data are to low for densities, this must be RAI or activity or whatever!)
Fig3: In Figure the signs are replaced by brackets! Please correct!d

·

Basic reporting

This is a well written study that is using camera trap data to infer spatio-tempral variation in activity of wild boars. I found the text very well constructed, and the analyses mostly sound. I have to major issues with the manuscript, that I think warrant some attention:

1. All presentations of the results, and the following discussion about them, is given in present tense. This causes the text to vastly over state the significance of the study. All presentations of the results, and all discussions of the results in the study, needs to be given in past tense. This makes it more specific statements for this particular study, and not broad general statements which is not warranted by the data or analyses. Some examples (not exhaustive, all reporting of and discussion about the results need to be changed in this way): Line 228 "...wild boar display a bimodal..." needs to be changed to "wild boar displayed a bimodal...), Line 318 "In our study, wild boar do not seem to avoid..." needs to be changed to "In our study, wild boar did not seem to have avoided...", and so on.

2. The authors have used a frequency based GAM to model the activity. This is not wrong in itself, but it goes against the vast majority of recent literature that is doing the same types of analyses with the same type of data. These studies almost exclusive make use of occupancy models, which accounts of imperfect detection by incorporating a detection function in the model. Here, lack of detection is regarded as absolute. I have two things to request of the authors here. First, I would like to see a justification for why the authors choose to simplify the model to exclude a detection function. Second, I would really like to see a discussion about possible benefits and draw-back of their approach in relation to others that have been developed for these specific types of analyses, perhaps specifically hierarchical occupancy models fitted under a Bayesian framework.

Experimental design

I think the experimental setup is good, but I would perhaps like to seem some justification for i. the decision to not place cameras on trails, and ii. to use only 2 min between successive observations as a limit for temporal independence.

Regarding the GAM, I would also like to see a better referencing to other methodologies that has been used for the same purpose, and a discussion about the pros and cons of the authors solution to this problem (i.e. estimation of activity simultaneous across both spatial and temporal dimensions).

Validity of the findings

The findings are valid, but the reporting of them not. All reporting of results should be changed into past tense, as well as any discussion about the results

Additional comments

Some minor comments and suggestions:

Lines 57-58: This sentence seem to indicate that this is a novel question, which couldn't be further from the truth. I think there are some missing references here regarding which methodology has been used previously for these types of questions.

Lines 87-90: This is a very vague statement. I suggest that it is either removed or much better focused in describing how the authors believe their study could aid the management of this species or system.

Line 101: I presume it is mean annual rainfall, which I think should be written out.

Line 123: Camera placement is a very complex issue. I found this criteria potentially concerning, since my own experiences of wild boar (and a large range of other species) is that they mainly travel along paths and trails. Hence, I see no reason for this, and I think it could possibly bias the data. I would like to see a brief justification for the decision to not place the cameras along trails, and also some possible implications of this placements on the data.

Line 125: This is a another contentious issue. For a group living species, I would generally find 2 min to be way too short as a demarcation of independent observations. However, recent studies have suggested that such demarcations should be omitted (Peral et al. 2022, Ecol Evol, 12:e9408). Hence, I would like to see some kind of justification for why such a short limit was used.

Line 136: Please describe what a "high seat" is.

Lines 175-179: Please give the link function.

Lines 226-258: As indicated earlier, all results should be presented in past tense.

Line 273: After the comma, what is depending on the context? Something is missing in this phrase.

Line 275: Remove the bracket and the text inside it.

Line 277: tends - > tended

Line 318 and 325: do not -> did not

Lines 350-359: Here I am missing a discussion about hierarchical occupancy models. Since they are so common, I think they need to be mentioned somewhere.

Line 375: adjust -> adjusted

---

## Round 0.2 · Major Revisions

Fortunately, both previous reviewers were available to review the revised manuscript. Unfortunately, they had very different opinions regarding the you changes made. Reviewer 1 concluded that the changes were appropriate and complete and recommended publication. Reviewer 2, in contrast, indicated that major problems in both content and form remained and recommended rejection; in the comments to the editor, however, the reviewer indicated that a re-submission could be considered making this a recommendation for major revisions rather than outright rejection. My reading indicated that my major concerns and those of the reviewers had been met, but that numerous small errors of grammar and word use remained requiring minor revisions. However, I do not have the expertise to evaluate the statistical issues and the reviewer’s concerns need to be considered carefully.

Reviewer 2 identified two major issues that could require reanalysis. The first is a problem of pseudoreplication resulting from considering records as independent when separated by gaps of 2 min or more, when in fact they result from movements of the same individuals. When I read your manuscript, it seemed to me that you were justifying your decision of this short criterion based on the need for more data points (L161). Of course, it is never justified to split autocorrelated data based simply on the need for a larger sample size. However, when I checked the paper you provided to support your decision (Peral et al., 2022), I realized I had misunderstood your point. Peral et al. article showed that the criterion for independent observations can affect conclusions about activity patterns. Their recommendation was not use long criteria, despite the concern about pseudoreplication which they downplayed on the basis of the importance of continuous records for activity research. However, the article did not really address the question about which pattern is the ‘real’ one and is too recent to have received responses from the larger community of activity pattern researchers. I recommend that you consult with a statistician to examine the validity of the argument by Peral et al. If you conclude that the argument is valid, I will accept the manuscript but it will be important that you include a more complete explanation of the argument in favor of this short criterion. If you and the statistician conclude that Peral et al. do not have a valid argument, you will need to redo your analysis with a more appropriate criterion.

The second concern of Reviewer 2 is the potential effect of including data from the Covid lockdown period on the final conclusions. You use the increased number of hikers during the Covid period as a possible explanation for a period of low activity, but do not directly address the appropriateness of including data from this period in your analysis. It does seem to me that, at a minimum, running the analysis without the lockdown period to see whether the pattern changes would be appropriate.

A third issue raised by Reviewer 2 is that it would have been useful to have included data from the period with more active hunting in the analysis. I can see that for the effect of hunting, that would make sense. On the other hand, for the effect of hunting on use of agricultural fields during the growing season, it would not be directly relevant. While I would not oppose this if you decide that the reviewer’s comment makes sense, I do not feel that it is required.

Regarding the reviewer’s suggestion that you reduce the second focus on the use of GAMMs or increase the coverage of alternative methods, I do not agree. I feel that the present balance between spatio-temporal patterns in boar activity and use of an alternative statistical approach is adequate. It seems fairly common for a researcher to use a newer method and show its strengths and weaknesses in an analysis of real data asking real questions.

Finally, I have again provided a pdf with highlights to indicate problems and inserted comments to suggest solutions.

Reviewer 1 ·

Basic reporting

All standards of thh journal ar fit!

Experimental design

The experimental design fits the journals aims

Validity of the findings

Findings are well discussed and concluded.

Additional comments

all recommended changes are included. The the MS fits now to the requirements.
Well done
congrats
Oliver

·

Basic reporting

I found much of the text in this revised version less clear than in the previous one. Although the language is partly acceptable, it is riddled with errors. In addition, I found the syntax at times overly complex, and much of the content rather unstructured. In particular, as noted in one of my general points raised in the "additional comments", I found the focus of the manuscript to be rather unclear. Hence, careful revision of the text would be necessary, both in terms of form and content.

Experimental design

Following revisions, I have some serious issues with the experimental design. I have three key concerns, each outlined in detail in the "additional comments" section:

- The data is (deliberately?) pseudoreplicated, and I do not think the independent sample sizes may be enough for the complex models they authors are trying to fit. Hence, I recommend the authors to re-do the analyses incorporating the dependence structures among observations, or using some form of spatial or temporal filtering to remove them. This would likely mean that the data need to be pooled, e.g., across years, with subsequent model simplification (e.g., with the removal of year) as a consequence.

- I find it highly in-appropriate to combine animal activity data from periods of Covid-19 lock down with data where humans were allowed to move around freely in the same model. Hence, I recommend the data is re-analysed with the Covid-19 lock down periods removed.

- The study system presents an interesting range of spatial contrasts in hunting regimes, but the data has been seasonally filtered to not take full advantage of these contrasts. I therefore suggest that data are included so that the full range of hunting regimes can be evaluated.

Validity of the findings

As indicated above, I have some issues regarding sample sizes of independent observations as well as with the selection of the data. I therefore have some doubts that the models have produced unbiased predictions.

Additional comments

This is fairly rare, but I must confess that I feel less comfortable with this revised version compared to the initial one that I reviewed. I think the key reason for this is that the revisions have clarified some issues related to the methodology, which now stands as rather questionable, at least to me given the data that has been used. In addition, following revisions, the key focus of the manuscript has been lost. It now stands half way between being a methodological evaluation of using GAMs to address time-space questions based on camera traps and a study on a specific wildlife management system which has used GAMs as the selected analytical method. I have the following large concerns and recommendations for this version of the manuscript.

- The key focus of the manuscript is not clear. The initial paragraph of the introduction and sections of the conclusions (e.g., lines 478-482, 494-512), present the manuscript as a methodological evaluation of using GAMs as an alternative method to study time-space use in animal activity from camera trapping data, whereas the hypotheses (lines 114-119) and the initial sentence of the conclusions (lines477-478) clearly focus the manuscript as an ecological study of how humans influence temporal and spatial patterns of activity in this mammal species. I strongly recommend the authors to take the manuscript more clearly along one of these two contrasting directions. Personally, I think toning down the methodological focus may be easier. If a methodological focus is kept, a much more rigid evaluation of the GAM approach, in comparison to alternative approaches based on camera data is absolutely necessary (e.g., the rapid development of hierarchical occupancy models fitted using a Bayesian paradigm is not even mentioned in the manuscript, despite them being the most obvious alternative to the GAM approach presented here). If a biological focus is enhanced, I think the hypotheses need to be more directly justified by a clearly identified research question. At present, I am not sure why we would be interested in if hunters influence diel variations in the space use of wild boars. This needs to be clarified in a revised version following this focus. This would mean that the structure of the introduction may need to be revised.

- Following some of the responses to the review questions, I am starting to have some very serious concerns with the adopted analytical approach, as well as with the data. For instance, the study seems to be deliberately pseudo-replicated. Although the random effect structure is not at all clear (i.e., lines 236-239), I assume that animal or group ID is not among the random terms. Hence, keeping a 2 minute time difference for a model that is fitted under a maximum likelihood framework, which assumes independent observations, is very questionable indeed. One of the key reasons for the relatively few implementations of analyses that estimate animal activity simultaneously across spatial and temporal dimensions is that they require very large sample sizes. Deliberately pseudo-replicating the data is not a very satisfactory solution to this problem. I also do not fully see how GAMs would be any better than other approaches for overcoming this issue. I would recommend the authors to re-do the analyses with a more stringent criteria for observational independence, or to incorporate the dependence structures of the data into the models (e.g., as appropriate random effect structures or co-variance matrices). They also need to provide a firm justification for the method chosen. The current one, for instance, to refer to a methodological study that was based not based on a likelihood algorithm (the cited study of Peral et al. 2022 used a non-parametric Kernel estimator, not a likelihood based regression model) is is not very acceptable. Bottom line here is that I think the sample sizes of independent observations likely are not enough for the relatively complex models he authors are trying to fit, and I would highly recommend that that the models are re-fitted using some more stringent criteria for observational independence, which probably would require the data to be pooled across years to give viable sample sizes.

- Apart from the non-independence, I think the selection of the data may need some revisions. First, I got very surprised to see that data form the Covid-19 lock down was included in the manuscript (lines 379-381). I think data form this period, for obvious reasons, need to be removed, or at least treated separately from other data. I am also really confused as to why the data is restricted to months where only part of the full spatial contrast in hunting regimes were active (lines 137-144). I do not see any reason for this restriction, and it certainly have not been justified in the manuscript. Hence, I would recommend that the analyses are re-done with any data from Covid-19 lock-down periods removed, and using a seasonal filter that allow for the full set of hunting regimes to be active, or at least provide a firm explanation to why the study has been focused purely on the summer.

---

## Round 0.3 · accepted · Accept

The authors have carefully addressed criticisms from the reviewer and editor. The article is now much clearer and more readible. While problems still remain, they are now clearly laid out for the readers.